# Association between *Interleukin-1* Polymorphisms and Susceptibility to Dental Peri-Implant Disease: A Meta-Analysis

**DOI:** 10.3390/pathogens10121600

**Published:** 2021-12-09

**Authors:** Hady Mohammadi, Mehrnoush Momeni Roochi, Masoud Sadeghi, Ata Garajei, Hosein Heidar, Ali Aghaie Meybodi, Mohsen Dallband, Sarton Mostafavi, Melina Mostafavi, Mojtaba Salehi, Jyothi Tadakamadla, Dena Sadeghi-Bahmani, Serge Brand

**Affiliations:** 1Department of Oral and Maxillofacial Surgery, Fellowship in Maxillofacial Trauma, Health Services, Kurdistan University of Medical Sciences, Sanandaj 6617713446, Iran; hadi.mohammadi@muk.ac.ir; 2Department of Oral and Maxillofacial Surgery, Fellowship in Maxillofacial Trauma, School of Dentistry, Tehran University of Medical Sciences, Tehran 1439955991, Iran; mehrnoushmomeni@yahoo.com (M.M.R.); drheidar@yahoo.com (H.H.); mojtabasalehi9241@gmail.com (M.S.); 3Department of Biology, Science and Research Branch, Islamic Azad University, Tehran 1477893855, Iran; sadeghi_mbrc@yahoo.com; 4Department of Head and Neck Surgical Oncology and Reconstructive Surgery, The Cancer Institute, School of Medicine, Tehran University of Medical Sciences, Tehran 1439955991, Iran; atagarajei@tums.ac.ir (A.G.); ali.aghaimeybodi@gmail.com (A.A.M.); 5Department of Oral and Maxillofacial Surgery, Dental School, Taleghani Hospital, Shahid Beheshti University of Medical Sciences, Tehran 1983963113, Iran; m-dalband@sbmu.ac.ir; 6English Department, Baneh Branch, Islamic Azad University, Baneh 6691133845, Iran; sartonmostafave@yahoo.com; 7Tehran Medical Branch, Islamic Azad University, Tehran 1419733171, Iran; mostafavi.melina@yahoo.com; 8School of Medicine and Dentistry, Griffith University, Brisbane, QLD 4222, Australia; j.tadakamadla@griffith.edu.au; 9Sleep Disorders Research Center, Kermanshah University of Medical Sciences, Kermanshah 6719851115, Iran; bahmanid@stanford.edu; 10Center for Affective, Stress and Sleep Disorders, University of Basel, Psychiatric Clinics, 4001 Basel, Switzerland; 11Department of Psychology, Stanford University, Stanford, CA 94305, USA; 12Substance Abuse Prevention Research Center, Kermanshah University of Medical Sciences, Kermanshah 6715847141, Iran; 13Division of Sport Science and Psychosocial Health, Department of Sport, Exercise and Health, University of Basel, 4052 Basel, Switzerland; 14School of Medicine, Tehran University of Medical Sciences, Tehran 1416753955, Iran

**Keywords:** peri-implant disease, peri-implantitis, bone loss, polymorphism, *interleukin−1*

## Abstract

Background and objective: Interleukins (ILs), as important biochemical mediators, control the host response to inflammation and are associated with bone resorption. In the present meta-analysis, we investigated the association between *IL−1* polymorphisms and susceptibility to dental peri-implant disease (PID). Materials and methods: We searched Web of Science, Cochrane Library, Scopus, and PubMed/Medline databases for studies published until 9 September2021, without any restrictions. We calculated the crude OR and 95% confidence intervals (CI) to estimate the associations between *IL−1* polymorphisms and PID risk in the five genetic models. We further performed the subgroup analysis, sensitivity analysis, meta-regression, trial sequential analysis, and calculated the publication bias. Results: Out of 212 retrieved records, sixteen articles were used in the meta-analysis. There was no association between *IL−1A (–889), IL−1B (−511), IL−1B (+3953),* and *IL−1RN (VNTR)* polymorphisms and the risk of dental PIDs, but there was an increased risk of *IL−1B (+3954)* in the patients with PIDs. In addition, an association of the composite genotype of *IL−1A (−889)/IL−1B (+3953)* was observed with the risk of PIDs, but not for the composite genotype of *IL−1A (−889)/IL−1B (+3954)*. The publication year, the ethnicity, sample size, and the outcome were significantly influenced pooled estimates of some genetic models. Trial sequential analysis showed the lack of sufficient sample sizes in the studies. **Conclusions**: Among IL−1 polymorphisms evaluated in the meta-analysis, the composite genotype of *IL−1A (−889)/IL−1B (+3953)* and *IL−1B (+3954)* were the only polymorphisms associated with the risk of PID. The T allele and CT genotype of *IL−1B (+3954)* polymorphism were also associated with an elevated risk of PID.

## 1. Introduction

Dental implants are currently considered an effective treatment for functional and cosmetic rehabilitation of patients with partial or complete edentulousness [1,2]. The clinical success of dental implants is based on the principle of osseointegration, which involves bone growth in metal implants. Multiple factors, including biological [3], may affect the success of the osseointegration. Peri-implantitis can lead to bone loss and finally implant failure [4,5]. Peri-implantitis, marginal bone loss, and implant failure are three outcomes associated with peri-implant diseases (PIDs) [6,7]. PID is a collective term for reversible peri-implant mucositis and irreversible peri-implantitis [8]. Peri-implantitis could negatively affect the quality of life [9].

In this view, meta-analyses [6,7,8,9,10], reviews [11], and original articles [12,13] demonstrated the role of several polymorphisms in PIDs. Proinflammatory cytokines, such as interleukins (ILs), are important biochemical mediators to control the host response to inflammation and to also stimulate the production and secretion of prostaglandins. Prostaglandins are associated with bone resorption and the metalloproteinases, which are involved in collagen degradation [14]. As such, IL−1 may be a useful indicator and biomarker in diagnosing peri-implantitis, especially because it has an important role in the periodontitis pathogenesis, and because it interferes with immune and inflammation processes, tissue damage, and homeostasis [15]. IL−1 is composed of 11 genes in the 430-kb fragment in the long arm DNA of chromosome 2, in the 2q12-q21 region. These genes produce the IL−1 alpha (IL−1A) and IL−1 beta (IL−1B) with genetic and biochemical differences but with the same biological functions [10,11,12,13,14,15,16]. *IL−1 receptor antagonist* (*IL−1RN*) gene regulates the synthesis of the IL−1ra antagonist protein, which can disrupt IL−1A and IL−1B function in competition for receptor binding [17].

A thorough literature search identified five systematic reviews [3,4,5,6,7,8,9,10,11,12,13,14,15,16,17,18,19,20] and three meta-analyses [10,11,12,13,14,15,16,17,18,19,20,21,22] focusing on the associations between IL−1 polymorphisms and PIDs. Among these meta-analyses, one meta-analysis [10] included the highest number of articles (13 articles) and reported an association between the occurrence of *IL−1A (−889)*, *IL−1B (−511)*, and *IL−1B (+3954)* polymorphisms in patients with PIDs. In contrast, the meta-analysis [10] reported no subgroup analysis, meta-regression, or trial sequential analysis (TSA); further, *IL−1B (+3954)* and *IL−1B (+3954)* polymorphisms were entered in the analyses without further distinctions, and last, the meta-analysis [10] included studies with a deviation from the Hardy–Weinberg equilibrium (HWE) in their control groups, along with studies with sample sizes with less than 10 cases. To counter this, the present meta-analysis expanded upon previous meta-analyses in three ways. First, the number of included studies was higher. Second, the number of statistical procedures was higher, and the statistical procedures were more complex and sophisticated. More specifically, to counterbalance possible biases and heterogeneity in the results, we employed procedures such as meta-regression and trial sequential analysis (TSA). Third and relatedly, we deleted those studies, in the event that in their control conditions a deviation from the Hardy–Weinberg equilibrium (HWE) could be observed. Given this background, the aims of the present comprehensive meta-analyses were as follows: to evaluate the association of *1A (−889)*, *IL−1B (−511)*, and *IL−1B (+3954)*, *IL−1B (+3954)*, and *IL−1RN (VNTR)* polymorphisms with PIDs; and to conduct subgroup analysis, meta-regression, and TSA. To this end, we removed those studies with a deviation from HWE, separately analyzed *IL−1B (+3954)* and *IL−1B (+3954)* polymorphisms, and we considered only studies with a minimum of 10 cases.

## 2. Materials and Methods

### 2.1. Study Design

The guidelines of PRISMA were followed while reporting this meta-analysis [23]. The PECO (population, exposer, comparison, and outcomes) question [24] was: are *IL−1* polymorphisms associated with PID risk among people with dental implants?

### 2.2. Search Strategy

One author (M.S.) extracted the specific studies from the databases, and the same author removed duplicates and irrelevant studies.

The Web of Science, Cochrane Library, Scopus, and PubMed/Medline databases were searched for studies published until 9 September 2021, without any restrictions. The searched terms were:

(“oral implant*” or “dental implant*” or “peri-implant disease*” or “implant failure” or “implant loss” or “peri-implant” or “peri-implantitis” or “failing implant” or “implant bone loss”) and (“interleukin*” or “interleukin-1*” or “IL−1*” or “IL1*”) and (“variant*” or “polymorphism*” or “allele” or “genotype*”). In addition, we searched several sources (Google Scholar, Free Medical Journals, Library Genesis, and Science Direct) to retrieve relevant missed articles.

### 2.3. Inclusion and Exclusion Criteria

Inclusion criteria were (1) case-control studies; (2) dental PID was the outcome of interest; (3) studies reporting *IL−1A (−889)*, *IL−1B (−511); IL−1B (+3953); IL−1B (+3954);*
*IL−1RN (VNTR)*, and composite genotype of *IL−1A (−889)/IL−1B (+3953)* and *IL−1A (−889)/IL−1B (+3954)* polymorphisms; (4) studies with the required data to calculate the odds ratios (ORs) with 95% confidence intervals (CIs) for genetic models; and (5) studies with no deviation of HWE in their control groups.

Exclusion criteria were (1) studies without the required data regarding genotype distributions, (2) animal studies, meta-analyses, review articles, book chapters, and letters to the editors; and (3) studies including less than 10 cases in each group (case and control groups).

The second author (H.M.) screened all the titles and abstracts based on the eligibility criteria and included/excluded studies for full-text review. Another author (D.S.B.) re-checked the relevant articles. In the event of low agreement, a third reviewer (S.B.) took a final solution.

### 2.4. Data Extraction

One author (M.S.) independently extracted the information or data from each study and another author (J.T.) rechecked them. If there was a disagreement between the authors, a third author (H.M) took the final decision.

### 2.5. Quality of Assessment

Two authors (M.S. and H.M.) independently evaluated the quality of each included article using the modified Newcastle-Ottawa Scale (NOS) questionnaire (a maximum total score of 9 was possible for each study) [25].

### 2.6. Statistical Analyses

We used Review Manager 5.3 (RevMan 5.3) to calculate crude OR and 95% confidence intervals (CI) as an estimate of the association between *IL−1* polymorphisms and PID risk in the five genetic models. To assess the pooled OR significance, the Z-test was applied with a *p* < 0.05. The I^2^ statistic showed the heterogeneity, we used the random-effect model, if there was a statistically significant heterogeneity (*p* < 0.1 or I^2^ > 50%); if there was no significant heterogeneity, the fixed-effect model was used. Ethnicity, PID outcome, and sample size were criteria for subgroup analyses.

We used Chi-square tests to calculate the *p*-value of the HWE in the control group of each study; in such cases, a *p* < 0.05 was considered as a deviation from the HWE.

We used Egger’s and Begg’s tests to plot and analyze the funnel plots; if a *p* < 0.05, then this was interpreted as publication bias. To evaluate the stability of pooled data, we used sensitivity analyses (“one study removed” and “cumulative analysis”). We used the Comprehensive Meta-Analysis version 2.0 (CMA 2.0) to calculate publication bias tests and sensitivity analyses.

We performed a meta-regression to survey the impact of publication year, ethnicity, PID outcome, and sample size on the pooled results. We used SPSS^®^ version 22.0 (IBM Corporation, Armonk, NY; USA) to perform the meta-regression.

To conduct the trial sequential analysis (TSA) we used the TSA software (version 0.9.5.10 beta) (Copenhagen Trial Unit, Centre for Clinical Intervention Research, Rigshospitalet, Copenhagen, Denmark). Running TSAs reduce these statistical errors [26], because each meta-analysis may create a false-positive or negative conclusion [27]. Based on an alpha risk of 5%, a beta risk of 20%, and a two-sided boundary type we computed the required information size (RIS). Studies were considered to have adequate sample sizes and lead to valid results, if the analyses of the Z-curve reached the RIS line, or monitored the boundary line or futility area. Otherwise, the amount of information was considered not to be large enough, suggesting the need for more evidence. A threshold of futility area showed no effect before reaching the information size.

## 3. Results

### 3.1. Study Selection

“Of the 212 papers retrieved in the databases, 112 were duplicates, and thus removed, leaving 100 titles and the abstracts for further evaluation. It turned out that 61 papers were irrelevant records, and thus excluded. At the end, 39 full-text articles were evaluated for eligibility” (Figure 1). Then, 23 articles were excluded with reasons (5 were systematic reviews, 3 were meta-analyses, 5 were reviews, 2 had no control groups, 1 was book chapter, 2 included less than 10 cases, 1 had a control group demonstrating HWE deviation, and 4 had insufficient data to estimate the odds ratios). At last, 16 articles were entered in the analysis.

### 3.2. Characteristics of the Studies

Appendix A provides the characteristics of sixteen articles [16,17,18,19,20,21,22,23,24,25,26,27,28,29,30,31,32,33,34,35,36,37,38,39,40,41,42] included in the meta-analysis. Nine articles [16,17,18,19,20,21,22,23,24,25,26,27,28,29,30,31,32,33,34,35,36,37,38,39,40,41,42] included Caucasians, three [29,30,31,32,33,34,35,36,37,38,39,40,41] Asians, and four [30,31,32,33,34,35,36,37] had participations with mixed ethnicity. Eight studies reported *IL−1A (−889)* polymorphism [16,28,29,30,31,37,39,41], eight [16,17,18,19,20,21,22,23,24,25,26,27,28,29,30,31,32,33,34,35,36,37,38,39,40,41] *IL−1B (−511)*, three [28,30,39] *IL−1B (+3953)* polymorphism, seven [16,17,18,19,20,21,22,23,24,25,26,27,28,29,30,31,32,33,34,35,36,37,38,39,40,41,42] *IL−1B (+3954)*, two [30,31,32,33,34,35,36,37,38,39,40] *IL−1RN (VNTR)* polymorphism, three [28,29,30,31,32,33,34,35,36,37,38] composite genotype of *IL−1A (−889)* and *IL−1B (+3953)*, and three [31,32,33,34,35,36] composite genotype of *IL−1A (−889)* and *IL−1B (+3954)* polymorphisms. Of the 16 articles, the outcomes were as follows: seven articles [16,17,18,19,20,21,22,23,24,25,26,27,28,29,30,31,32,33,34,35,36,37,38,39]: implant failure; six articles [31,32,33,34,35,36,37,38,39,40,41,42]: peri-implantitis; three articles [29,30,31,32,33,34,35,36,37,38,39,40,41]: marginal bone loss. Appendix A provides the quality scores of each study: 14 out of 16 studies were considered to be of high quality (score ≥ 7).

### 3.3. Pooled Analyses

As Table 1 shows, the pooled ORs for the association between alleles and genotypes of *IL−1A (−889)* polymorphism and the risk of dental PID were 1.19 (95% CI: 0.92, 1.55; *p* = 0.19; I^2^ = 0%) for allelic, 1.18 (95% CI: 0.62, 2.55; *p* = 0.61; I^2^ = 0%) for homozygous, 1.45 (95% CI: 0.97, 2.16; *p* = 0.07; I^2^ = 0%) for heterozygous, 1.43 (95% CI: 0.98, 2.10; *p* = 0.07; I^2^ = 0%) for recessive, and 1.02 (95% CI: 0.64, 1.63; *p* = 0.94; I^2^ = 0%) for dominant models. There was no association between *IL−1A (−889)* polymorphism and the risk of dental PID.

Table 2 shows the pooled analyses for the association between alleles and genotypes of *IL−1B (−511)* polymorphism and the dental PID risk. The pooled ORs were 1.10 (95% CI: 0.69, 1.75; *p* = 0.70; I^2^ = 75%) for allelic, 1.20 (95% CI: 0.59, 2.42; *p* = 0.61; I^2^ = 54%) for homozygous, 1.72 (95% CI: 0.52, 1.01; *p* = 0.06; I^2^ = 18%) for heterozygous, 1.84 (95% CI: 0.61, 1.14; *p* = 0.25; I^2^ = 48%) for recessive, and 1.45 (95% CI: 1.00, 2.09; *p* = 0.05; I^2^ = 42%) for dominant models. There was no association between *IL−1B (−511)* polymorphism and the risk of dental PID.

The pooled analyses for the association between alleles and genotypes of *IL−1B (+3953)* polymorphism and the dental PID risk are shown in Table 3. The pooled ORs were 1.45 (95%CI: 0.93, 2.27; *p* = 0.10; I^2^ = 0%) for allelic, 2.03 (95% CI: 0.54, 7.55; *p* = 0.29; I^2^ = 0%) for homozygous, 1.53 (95% CI: 0.87, 2.70; *p* = 0.14; I^2^ = 0%) for heterozygous, 1.58 (95% CI: 0.91, 2.74; *p* = 0.10; I^2^ = 0%) for recessive, and 1.78 (95% CI: 0.49, 6.42; *p* = 0.38; I^2^ = 0%) for dominant models. There was no association between *IL−1B (+3953)* polymorphism and the risk of dental PID.

The pooled analyses for the association between alleles and genotypes of *IL−1B (+3954)* polymorphism and the dental PID risk are illustrated in Table 4. The pooled ORs were 2.04 (95% CI: 1.02, 4.08; *p* = 0.04; I^2^ = 73%) for allelic, 1.73 (95% CI: 0.45, 6.58; *p* = 0.42; I^2^ = 52%) for homozygous, 1.68 (95%CI: 0.18, 2.39; *p* = 0.004; I^2^ = 49%) for heterozygous, 2.27 (95% CI: 1.11, 4.64; *p* = 0.03; I^2^ = 65%) for recessive, and 1.19 (95% CI: 0.58, 2.45; *p* = 0.63; I^2^ = 42%) for dominant models. Significant associations were observed between the T allele, CT genotype of *IL−1B (+3954)* polymorphism, and the susceptibility to dental PID.

The pooled analyses for the association between alleles and genotypes of *IL−1RN (VNTR)* polymorphism and the dental PID susceptibility are shown in Table 5. The pooled ORs were 0.91 (95% CI: 0.34, 2.49; *p* = 0.86; I^2^ = 74%), 0.40 (95% CI: 0.01, 23.82; *p* = 0.66; I^2^ = 82%), 0.86 (95% CI: 0.26, 2.81; *p* = 0.80; I^2^ = 66%), 0.81 (95% CI: 0.19, 3.45; *p* = 0.77; I^2^ = 79%), and 0.74 (95% CI: 0.26, 2.09; *p* = 0.57; I^2^ = 32%) for allelic, homozygous, heterozygous, recessive, and dominant models, respectively. There was no association between *IL−1RN (VNTR)* polymorphism and the risk of dental PID.

Table 6 reports the pooled analyses for the association of the composite genotype of *IL−1A (−889)/IL−1B (+3953)* and *IL−1A (−889)/IL−1B (+3954)* with the risk of dental PID (genotype-positive vs. -negative). The pooled ORs were 1.73 (95% CI: 1.03, 2.92; *p* = 0.04; I^2^ = 0%) for the composite genotype of *IL−1A (−889)/IL−1B (+3953)* and 2.31 (95% CI: 0.65, 8.17; *p* = 0.20; I^2^ = 82%), 0.86 (95% CI: 0.26, 2.81; *p* = 0.80; I^2^ = 66%), 0.81 (95% CI: 0.19, 3.45; *p* = 0.77; I^2^ = 72%) for the composite genotype of *IL−1A (−889)/IL−1B (+3954)*. An association was only observed between the composite genotype of *IL−1A (−889)/IL−1B (+3953)* and the risk of dental PID.

### 3.4. Subgroup Analysis

The subgroup analyses (based on the ethnicity, PID outcome, and the sample size) of the association between *IL−1A (−889)* polymorphism and the risk of dental PID are shown in Appendix A. The results showed that the ethnicity and the outcome were two significant factors that could affect the pooled estimates for the association between alleles, genotypes of *IL−1A (−889)* polymorphism and the risk of dental PID in heterozygous and recessive models.

The subgroup analyses (the ethnicity, the outcome, and the sample size) of the association between *IL−1B (−511)* polymorphism and the dental PID risk are shown in Appendix A. The findings reveal that the ethnicity, the outcome, and the sample size were significant factors influencing the pooled analysis for the association between of *IL−1B (−511)* polymorphism and the risk of dental PID in the heterozygous model. For the dominant model, ethnicity, PID outcome, and sample size influenced the pooled estimates.

Appendix A shows the subgroup analyses (the ethnicity, the outcome, and the sample size) of the association between *IL−1B (+3954)* polymorphism and the dental PID risk. The results suggest that sample size was a significant factor influencing the pooled analysis for the association between of *IL−1B (+3954)* polymorphism and the risk of dental PID in allelic, heterozygous, and dominant models.

### 3.5. Meta-Regression

Appendix A provides the results of the meta-regression analysis to evaluate the effect of publication year, sample size, ethnicity, and PID outcome on the association between *IL−1A (−889)*, *IL−1B (−511)*, and *IL−1B (+3954)* polymorphisms and the risk of dental PID. The publication year predicted pooled results, and just for the dominant model of *IL−1B (+3954)* polymorphism.

### 3.6. Sensitivity Analysis

Both “one study removed” and “cumulative analysis” were performed for the sensitivity analyses that included at least three studies; results remained stable.

### 3.7. Trial Sequential Analysis

Appendix A shows the TSA based on recessive model for the association of *IL−1A (−889)*, *IL−1B (−511)*, and *IL−1B (+3954)* polymorphisms with the risk of dental PID. The Z-curve did neither reach the RIS line, nor monitor the boundary line or futility area; as a result, there was inadequate evidence and therefore, more information was needed.

### 3.8. Publication Bias

We plotted the funnel plots (Appendix A) and calculated the *p*-values of Egger’s and Begg’s tests to evaluate the publication bias across the studies in the analyses that included at least three studies. There was no publication bias across the studies, except homozygous (*p*-value of Egger’s test: 0.007) and dominant models (*p*-value of Egger’s test: 0.013) of *IL−1B (+3954)* polymorphism.

## 4. Discussion

The main findings of the present meta-analysis showed that there was no association between *IL−1A (−889)*, *IL−1B (−511)*, *IL−1B (+3953)*, and *IL−1RN (VNTR)* polymorphisms and the risk of dental PIDs. In contrast, there was an increased risk of *IL−1B (+3954)* in the patients with PIDs. In addition, an association was observed between the composite genotype of *IL−1A (−889)/IL−1B (+3953)* and PIDs, but not between the composite genotype of *IL−1A (−889)/IL−1B (+3954)* and PIDs. Further, the subgroup analysis showed that ethnicity and PID outcomes influenced the association of *IL−1A (−889)* polymorphism and the risk of PID. Ethnicity, PID outcome, and sample sizes were significant factors for *IL−1B (−511)* polymorphism, while the sample size influenced the *IL−1B (+3954)* polymorphism. Further, based on meta-regression, the publication year was a significant predictor of the pooled results of *IL−1B (+3954)* polymorphism. Last, the TSA showed that there were inadequate sample sizes among the studies included in the analyses.

Three published meta-analyses [10,11,12,13,14,15,16,17,18,19,20,21,22] investigated the association between the *IL−1* polymorphisms and the risk of PIDs. Junior et al. [21] reported just two articles and reported that there was no association between *IL−1B (−511)* polymorphism and the risk of implant failure based on the allelic model. Liao et al. [10] included 13 articles reporting *IL−1A (−889)*, *IL−1B (−511)*, *IL−1B (+3954)*, and *IL−1RN (VNTR)* polymorphisms and the risk of dental PIDs and also mixing the patients with peri-implantitis, implant loss, and marginal bone loss based on the allelic model. This study [10] included both *IL−1B (+3953)* and *IL−1B (+3954)* in a similar analysis. The authors found that *IL−1B (−511)* polymorphism and the composite genotype of *IL−1A (−889)*/*IL−1B (+3954)* on risk for implant failure and peri-implantitis. Third, meta-analysis [22] included two articles to check the association of *IL−1* polymorphisms (*IL−1A (−889)*, *IL−1B (−511)*, and *IL−1B (+3954)*) with early crestal bone loss around submerged dental implants that there was just an association between *IL−1B (−511)* polymorphisms and early crestal bone loss. Our meta-analysis included 16 articles to investigate the association between the *IL−1* polymorphisms and the risk of PIDs. In addition, we included peri-implantitis, implant loss, and marginal bone loss as PIDs and mixed them in the first analysis such as the meta-analysis of Liao et al. [10], but in subgroup analysis, we separately analyzed them for each polymorphism. Unlike the previously mentioned meta-analyzes, we used five genetic models, meta-regression, and TSA, as well as removed studies with a deviation from HWE in their control group to reduce bias and heterogeneity.

*IL−1A (−889)* polymorphism was found to be related to chronic periodontal disease in Brazilian cases [43]. Further, for *IL−1A (−889)* polymorphism, T allele compared to C allele induced a four times higher expression of IL−1 alpha [44] and also TT genotypes compared to CC genotype [45]. Similarly, Cosyn et al. [16] presented the association between *IL−1A (−889)* polymorphism and the risk of implant failure. In contrast and unlike previous studies [28,29,30,31,32,33,34,35,36,37,38,39,40,41], we were unable to identify an association between this polymorphism and the risk of PID. However, our subgroup analysis showed that the association between *IL−1A (−889)* polymorphism and the risk of implant failure was statistically significant. Therefore, PID outcomes appeared to be important factors to explain the association between *IL−1A (−889)* polymorphism and PID risk.

*IL−1B (−511)* polymorphism is similar to *IL−1B (+3954)* polymorphism, which was found to have a strong role in chronic periodontitis and inflammation [46]. With regards to the association of *IL−1B (−511)* polymorphism with the risk of PID, one study [33] showed an elevated risk, while another study [41] reported a protective role of this polymorphism. In this view, the present meta-analysis was unable to confirm the association between *IL−1B (−511)* polymorphism and the risk of PID, and this zero association was already observed elsewhere [16,17,18,19,20,21,22,23,24,25,26,27,28,29,30,31,32,33,34,35,36,37]. However, the subgroup analysis showed that TC genotype has a protective role on marginal bone loss in Asian individuals. Therefore, the role of ethnicity should be considered when focusing on the association between *IL−1B* polymorphisms and the risk of PID.

The *IL−1* gene polymorphism may have a negative effect on the results of peri-implantitis treatment in genotype-positive individuals, and the combination of *IL−1A (−889)/IL−1B (+3954)* in peri-implant tissues may act as a risk factor that elevates tissue destruction [36]. In this view, polymorphisms may be involved in osseointegration through the cumulative effect of multiple polymorphisms [47]. In our meta-analysis, the pooled results showed that the combination of *IL−1A (−889)/IL−1B (+3953)* could act as a risk factor for PID, while this was not the case for the combination of *IL−1A (−889)/IL−1B (+3954).* Further, the prevalence of these combinations varied among ethnic groups [36]. Therefore, in future studies, and due to the different results between the combination of *IL−1* polymorphisms and the risk of PID, the combination of *IL−1* polymorphisms with emphasis on ethnicity demand special attention. In addition, the functional genetic polymorphisms of *IL−1B (+3954)* and *IL−1RN (VNTR)* may diversify the production of IL−1b and IL−1ra proteins [48,49]. *IL−1B* and *IL−1RA* may act as regulators of the inflammatory immune system [50]; as a result, polymorphisms in these genes can affect inflammation and cause implant failure [31,32]. Given this background, two studies reported that *IL−1B (+3954)* polymorphism could play a role in the pathogenesis of peri-implantitis and increase its risk [16,42]. In accordance, our meta-analysis confirmed the result that *IL−1B (+3954)* polymorphism caused an increased risk of PID, but not for individuals with peri-implantitis. In the same vein, our meta-analysis and several other individual studies [30,31,32,33,34,35,36,37,38,39,40] did not confirm the association of *IL−1RN (VNTR)* polymorphism with the risk of PID.

The success of dental implants is determined by several factors such as clinical, biomechanical, and genetic risk [51,52]. Further, the synergistic effect of smoking and the positive *IL−1* genotype significantly increase the risk of implant failure [17]. Regardless of the status of the *IL−1* genotype, smoking was associated with elevated peri-implant bone loss and implant failure [53,54]. In our meta-analysis, several studies did not report the smoking status, or data on smoking prevalence among case and control groups were not reported; as such, smoking status was not entered as a further factor in the present meta-analysis. However, future studies should consider the smoking status and its correlation with implant failure and the prevalence of *IL−1* polymorphisms.

Despite the new results, several limitations should be considered. First, based on TSA, there was a lack of sufficient sample sizes in the included studies. Second, only a very few studies were available for the two polymorphisms (*IL−1B (+3953)* and *IL−1RN (VNTR)*), given this, subgroup analyses and meta-regression analyses for these polymorphisms were not possible. Third, a high heterogeneity across the studies in several analyses was observed. Fourth, several confounding factors were observed in the pooled results.

In contrast, the strengths of the meta-analysis were first, the lack of publication bias across the studies in most analyses; second, the stability of the results; and third, studies with a deviation from HWE in their control group were removed.

## 5. Conclusions

The main findings of the meta-analysis showed that there was no association between *IL−1A (−889)*, *IL−1B (−511)*, *IL−1B (+3953)*, and *IL−1RN (VNTR)* polymorphisms, the composite genotype of *IL−1A (−889)/IL−1B (+3954)* and the risk of dental PID. In contrast, the composite genotype of *IL−1A (−889)/IL−1B (+3953)* and *IL−1B (+3954)* polymorphism was associated with an elevated risk for PID. Further, other factors such as the publication bias, ethnicity, PID outcome, and sample size affected the pooled results. In addition, small sample sizes and high heterogeneity across the studies showed that the power and accuracy of the results appeared to be low. Clinicians should pay attention to the effects of these polymorphisms on the outcomes of treatment. Given this, further larger and well-designed studies among people of different ethnicities and with detailed individual information (age, sex, and smoking status) are needed to confirm the present results.

## Figures and Tables

**Figure 1 pathogens-10-01600-f001:**
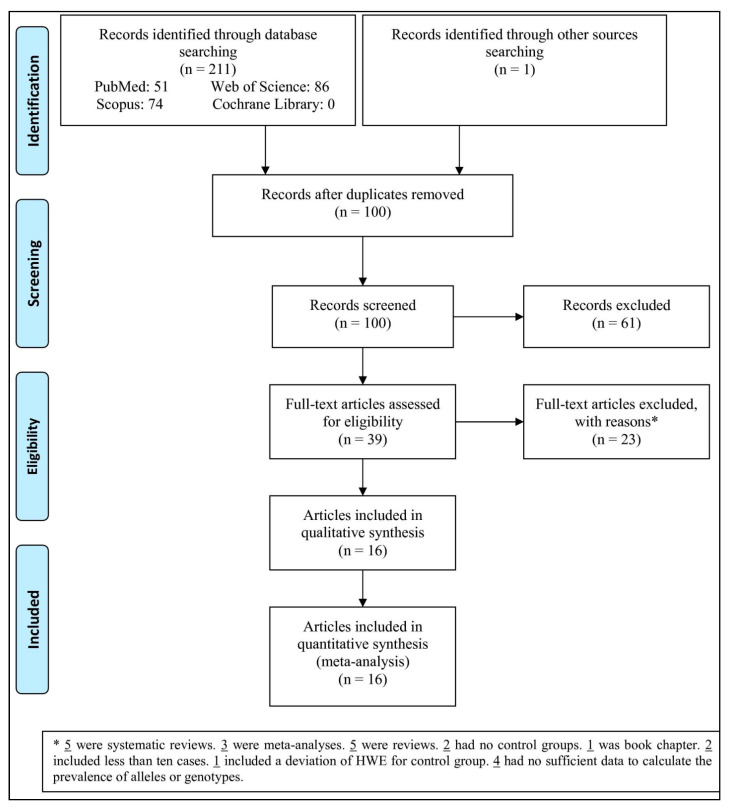
Flow chart of the study selection.

**Table 1 pathogens-10-01600-t001:** Pooled analysis of the association between alleles and genotypes of *IL−1A (−889)* polymorphism and the risk of dental PID.

Genetic Model	First Author, Publication Year	Case		Control		Weight	Odds Ratio
		Events	Total	Events	Total		M-H, Fixed, 95% CI
T vs. C	Rogers, 2002 [28]	10	38	16	62	8.8%	1.03 [0.41, 2.57]
	Shimpuku, 2003 [29]	4	34	4	44	3.0%	1.33 [0.31, 5.77]
	Campos, 2005 [30]	17	56	21	68	12.9%	0.98 [0.45, 2.10]
	Laine, 2006 [31]	49	142	33	98	25.0%	1.04 [0.60, 1.79]
	Melo, 2012 [37]	25	32	51	62	7.4%	0.77 [0.27, 2.23]
	Jacobi-Gresser, 2013 [39]	28	82	32	136	15.5%	1.69 [0.92, 3.08]
	Cosyn, 2016 [16]	10	24	4	26	2.2%	3.93 [1.03, 14.99]
	Agrawal, 2021 [41]	98	136	89	126	25.2%	1.07 [0.63, 1.83]
Subtotal (95% CI)			544		622	100.0%	1.19 [0.92, 1.55]
Total events		241		250			
Heterogeneity: Chi^2^ = 5.74, df = 7 (*p* = 0.57); I^2^ = 0% Test for overall effect: Z = 1.30 (*p* = 0.19)							
TT vs. CC	Rogers, 2002 [28]	0	9	3	21	12.2%	0.28 [0.01, 5.96]
	Shimpuku, 2003 [29]	1	15	0	18	2.4%	3.83 [0.14, 101.07]
	Campos, 2005 [30]	4	19	5	23	21.0%	0.96 [0.22, 4.23]
	Laine, 2006 [31]	2	26	3	22	17.6%	0.53 [0.08, 3.49]
	Melo, 2012 [37]	9	9	21	22	3.8%	1.33 [0.05, 35.60]
	Jacobi-Gresser, 2013 [39]	3	19	3	42	9.2%	2.44 [0.44, 13.38]
	Cosyn, 2016 [16]	2	6	0	9	1.6%	10.56 [0.41, 268.69]
	Agrawal, 2021 [41]	37	44	32	38	32.1%	0.99 [0.30, 3.25]
Subtotal (95% CI)			147		195	100.0%	1.18 [0.62, 2.25]
Total events		58		67			
Heterogeneity: Chi^2^ = 4.67, df = 7 (*p* = 0.70); I^2^ = 0% Test for overall effect: Z = 0.51 (*p* = 0.61)							
CT vs. CC	Rogers, 2002 [28]	10	19	10	28	9.6%	2.00 [0.61, 6.55]
	Shimpuku, 2003 [29]	2	16	4	22	7.4%	0.64 [0.10, 4.03]
	Campos, 2005 [30]	9	24	11	29	15.6%	0.98 [0.32, 3.00]
	Laine, 2006 [31]	45	69	27	46	28.3%	1.32 [0.61, 2.84]
	Melo, 2012 [37]	7	7	9	10	1.3%	2.37 [0.08, 66.88]
	Jacobi-Gresser, 2013 [39]	22	38	26	65	20.3%	2.06 [0.91, 4.65]
	Cosyn, 2016 [16]	6	10	4	13	3.5%	3.38 [0.60, 19.01]
	Agrawal, 2021 [41]	24	31	25	31	14.2%	0.82 [0.24, 2.80]
Subtotal (95% CI)			214		244	100.0%	1.45 [0.97, 2.16]
Total events			125		116		
Heterogeneity: Chi^2^ = 4.11, df = 7 (*p* = 0.77); I^2^ = 0% Test for overall effect: Z = 1.81 (*p* = 0.07)							
TT + CT vs. CC	Rogers, 2002 [28]	10	19	13	31	10.7%	1.54 [0.49, 4.85]
	Shimpuku, 2003 [29]	3	17	4	22	6.6%	0.96 [0.18, 5.03]
	Campos, 2005 [30]	13	28	16	34	17.7%	0.97 [0.36, 2.66]
	Laine, 2006 [31]	47	71	30	49	27.4%	1.24 [0.58, 2.64]
	Melo, 2012 [37]	16	16	30	31	1.4%	1.62 [0.06, 42.12]
	Jacobi-Gresser, 2013 [39]	25	41	29	68	19.4%	2.10 [0.95, 4.63]
	Cosyn, 2016 [16]	8	12	4	13	2.9%	4.50 [0.84, 24.18]
	Agrawal, 2021 [41]	61	68	57	63	13.9%	0.92 [0.29, 2.89]
Subtotal (95% CI)			272		311	100.0%	1.43 [0.98, 2.10]
Total events			183		183		
Heterogeneity: Chi^2^ = 4.21, df = 7 (*p* = 0.76); I^2^ = 0% Test for overall effect: Z = 1.83 (*p* = 0.07)							
TT vs. CC + CT	Rogers, 2002 [28]	0	19	3	31	7.7%	0.21 [0.01, 4.27]
	Shimpuku, 2003 [29]	1	17	0	22	1.2%	4.09 [0.16, 106.89]
	Campos, 2005 [30]	4	28	5	34	11.3%	0.97 [0.23, 4.01]
	Laine, 2006 [31]	2	71	3	49	10.1%	0.44 [0.07, 2.76]
	Melo, 2012 [37]	9	16	21	31	18.3%	0.61 [0.18, 2.12]
	Jacobi-Gresser, 2013 [39]	3	41	3	68	6.1%	1.71 [0.33, 8.90]
	Cosyn, 2016 [16]	2	12	0	13	1.1%	6.43 [0.28, 148.77]
	Agrawal, 2021 [41]	37	68	32	63	44.2%	1.16 [0.58, 2.30]
Subtotal (95% CI)			272		311	100.0%	1.02 [0.64, 1.63]
Total events			58		67		
Heterogeneity: Chi^2^ = 5.03, df = 7 (*p* = 0.66); I^2^ = 0% Test for overall effect: Z = 0.08 (*p* = 0.94)							

Abbreviation: CI: confidence interval; I^2^: heterogeneity.

**Table 2 pathogens-10-01600-t002:** Pooled analysis of the association between alleles and genotypes of *IL−1B (−511)* polymorphism and the risk of dental PID.

Genetic Model	First Author, Publication Year	Case		Control		Weight	Odds Ratio
		Events	Total	Events	Total		M-H, Random, 95% CI
T vs. C	Shimpuku, 2003 [29]	22	34	19	44	10.6%	2.41 [0.96, 6.07]
	Campos, 2005 [30]	21	56	30	68	12.6%	0.76 [0.37, 1.57]
	Laine, 2006 [31]	51	142	35	98	14.6%	1.01 [0.59, 1.73]
	Lin, 2007 [33]	37	58	19	60	12.2%	3.80 [1.77, 8.16]
	Dirschnabel, 2011 [35]	83	184	144	370	16.4%	1.29 [0.90, 1.84]
	Melo, 2012 [37]	25	32	51	62	9.4%	0.77 [0.27, 2.23]
	Cosyn, 2016 [16]	16	28	18	28	9.2%	0.74 [0.25, 2.17]
	Agrawal, 2021 [41]	40	136	61	126	14.9%	0.44 [0.27, 0.74]
Subtotal (95% CI)			670		856	100.0%	1.10 [0.69, 1.75]
Total events		295		377			
Heterogeneity: Tau^2^ = 0.32; Chi^2^ = 27.86, df = 7 (*p* = 0.0002); I^2^ = 75% Test for overall effect: Z = 0.38 (*p* = 0.70)							
TT vs. CC	Shimpuku, 2003 [29]	8	11	3	9	8.8%	5.33 [0.78, 36.33]
	Campos, 2005 [30]	7	21	7	18	13.4%	0.79 [0.21, 2.92]
	Laine, 2006 [31]	10	40	5	24	14.4%	1.27 [0.37, 4.28]
	Lin, 2007 [33]	14	20	6	15	12.6%	3.50 [0.86, 14.30]
	Dirschnabel, 2011 [35]	21	51	28	97	20.1%	1.73 [0.85, 3.51]
	Melo, 2012 [37]	2	6	3	13	7.6%	1.67 [0.20, 14.05]
	Cosyn, 2016 [16]	5	8	6	8	7.5%	0.56 [0.06, 4.76]
	Agrawal, 2021 [41]	7	42	13	28	15.6%	0.23 [0.08, 0.69]
Subtotal (95% CI)			199		212	100.0%	1.20 [0.59, 2.42]
Total events		74		71			
Heterogeneity: Tau^2^ = 0.51; Chi^2^ = 15.17, df = 7 (*p* = 0.03); I^2^ = 54% Test for overall effect: Z = 0.50 (*p* = 0.61)							
CT vs. CC	Shimpuku, 2003 [29]	6	9	13	19	3.4%	0.92 [0.17, 5.00]
	Campos, 2005 [30]	7	21	16	27	11.4%	0.34 [0.10, 1.13]
	Laine, 2006 [31]	31	61	25	44	17.5%	0.79 [0.36, 1.71]
	Lin, 2007 [33]	9	15	15	24	5.6%	0.90 [0.24, 3.38]
	Dirschnabel, 2011 [35]	41	71	88	157	28.3%	1.07 [0.61, 1.89]
	Melo, 2012 [37]	10	14	18	28	4.2%	1.39 [0.34, 5.60]
	Cosyn, 2016 [16]	6	9	6	8	2.6%	0.67 [0.08, 5.54]
	Agrawal, 2021 [41]	26	61	35	50	27.0%	0.32 [0.14, 0.70]
Subtotal (95% CI)			261		357	100.0%	0.72 [0.52, 1.01]
Total events		136		216			
Heterogeneity: Chi^2^ = 8.58, df = 7 (*p* = 0.28); I^2^ = 18% Test for overall effect: Z = 1.91 (*p* = 0.06)							
TT + CT vs. CC	Shimpuku, 2003 [29]	14	17	16	22	2.8%	1.75 [0.37, 8.33]
	Campos, 2005 [30]	14	28	23	34	11.7%	0.48 [0.17, 1.34]
	Laine, 2006 [31]	41	71	30	49	16.8%	0.87 [0.41, 1.82]
	Lin, 2007 [33]	23	29	21	30	4.8%	1.64 [0.50, 5.40]
	Dirschnabel, 2011 [35]	62	92	116	185	28.2%	1.23 [0.73, 2.08]
	Melo, 2012 [37]	12	16	21	31	4.0%	1.43 [0.37, 5.56]
	Cosyn, 2016 [16]	11	14	12	14	2.9%	0.61 [0.09, 4.37]
	Agrawal, 2021 [41]	33	68	48	63	28.8%	0.29 [0.14, 0.62]
Subtotal (95% CI)			335		428	100.0%	0.84 [0.61, 1.14]
Total events		210		287			
Heterogeneity: Chi^2^ = 13.40, df = 7 (*p* = 0.06); I^2^ = 48% Test for overall effect: Z = 1.15 (*p* = 0.25)							
TT vs. CC + CT	Shimpuku, 2003 [29]	8	17	3	22	3.0%	5.63 [1.20, 26.41]
	Campos, 2005 [30]	7	28	7	34	10.2%	1.29 [0.39, 4.24]
	Laine, 2006 [31]	10	71	5	49	11.0%	1.44 [0.46, 4.52]
	Lin, 2007 [33]	14	29	6	30	6.6%	3.73 [1.18, 11.83]
	Dirschnabel, 2011 [35]	21	92	28	185	31.0%	1.66 [0.88, 3.12]
	Melo, 2012 [37]	2	16	3	31	3.9%	1.33 [0.20, 8.92]
	Cosyn, 2016 [16]	5	14	6	14	8.3%	0.74 [0.16, 3.39]
	Agrawal, 2021 [41]	7	68	13	63	26.1%	0.44 [0.16, 1.19]
Subtotal (95% CI)			335		428	100.0%	1.45 [1.00, 2.09]
Total events		74		71			
Heterogeneity: Chi^2^ = 12.03, df = 7 (*p* = 0.10); I^2^ = 42% Test for overall effect: Z = 1.95 (*p* = 0.05)							

Abbreviation: CI: confidence interval; I^2^: heterogeneity. Fixed-effects model was used for homozygous (CT vs. CC), recessive (TT plus CT vs. CC), and dominant (TT vs. CC plus CT) models.

**Table 3 pathogens-10-01600-t003:** Pooled analysis of the association between alleles and genotypes of *IL−1B (+3953)* polymorphism and the risk of dental PID.

Genetic Model	First Author, Publication Year	Case		Control		Weight	Odds Ratio
		Events	Total	Events	Total		M-H, Fixed, 95% CI
T vs. C	Rogers, 2002 [28]	11	38	13	62	22.2%	1.54 [0.61, 3.89]
	Campos, 2005 [30]	13	56	14	68	30.7%	1.17 [0.50, 2.74]
	Jacobi-Gresser, 2013 [39]	24	82	28	136	47.1%	1.60 [0.85, 3.00]
Subtotal (95% CI)			176		266	100.0%	1.45 [0.93, 2.27]
Total events		48		55			
Heterogeneity: Chi^2^ = 0.35, df = 2 (*p* = 0.84); I^2^ = 0% Test for overall effect: Z = 1.64 (*p* = 0.10)							
TT vs. CC	Rogers, 2002 [28]	0	8	1	20	27.8%	0.76 [0.03, 20.74]
	Campos, 2005 [30]	3	21	2	24	52.4%	1.83 [0.28, 12.19]
	Jacobi-Gresser, 2013 [39]	2	21	1	42	19.8%	4.32 [0.37, 50.58]
Subtotal (95% CI)			50		86	100.0%	2.03 [0.54, 7.55]
Total events		5		4			
Heterogeneity: Chi^2^ = 0.71, df = 2 (*p* = 0.70); I^2^ = 0% Test for overall effect: Z = 1.05 (*p* = 0.29)							
CT vs. CC	Rogers, 2002 [28]	11	19	11	30	18.7%	2.38 [0.73, 7.69]
	Campos, 2005 [30]	7	25	10	32	32.8%	0.86 [0.27, 2.70]
	Jacobi-Gresser, 2013 [39]	20	39	26	67	48.5%	1.66 [0.75, 3.68]
Subtotal (95% CI)			83		129	100.0%	1.53 [0.87, 2.70]
Total events		38		47			
Heterogeneity: Chi^2^ = 1.56, df = 2 (*p* = 0.46); I^2^ = 0% Test for overall effect: Z = 1.47 (*p* = 0.14)							
TT + CT vs. CC	Rogers, 2002 [28]	11	19	12	31	19.0%	2.18 [0.68, 6.96]
	Campos, 2005 [30]	10	28	12	34	34.5%	1.02 [0.36, 2.90]
	Jacobi-Gresser, 2013 [39]	22	41	27	68	46.6%	1.76 [0.80, 3.85]
Subtotal (95% CI)			88		133	100.0%	1.58 [0.91, 2.74]
Total events		43		51			
Heterogeneity: Chi^2^ = 1.04, df = 2 (*p* = 0.59); I^2^ = 0% Test for overall effect: Z = 1.64 (*p* = 0.10)							
TT vs. CC + CT	Rogers, 2002 [28]	0	19	1	31	32.6%	0.52 [0.02, 13.46]
	Campos, 2005 [30]	3	28	2	34	46.7%	1.92 [0.30, 12.38]
	Jacobi-Gresser, 2013 [39]	2	41	1	68	20.7%	3.44 [0.30, 39.13]
Subtotal (95% CI)			88		133	100.0%	1.78 [0.49, 6.42]
Total events		5		4			
Heterogeneity: Chi^2^ = 0.84, df = 2 (*p* = 0.66); I^2^ = 0% Test for overall effect: Z = 0.88 (*p* = 0.38)							

Abbreviation: CI: confidence interval; I^2^: heterogeneity.

**Table 4 pathogens-10-01600-t004:** Pooled analysis of the association between alleles and genotypes of *IL−1B (+3954)* polymorphism and the risk of dental PID.

Genetic Model	First Author, Publication Year	Case		Control		Weight	Odds Ratio
		Events	Total	Events	Total		M-H, Random, 95% CI
T vs. C	Shimpuku, 2003 [29]	1	34	2	44	6.0%	0.64 [0.06, 7.33]
	Laine, 2006 [31]	40	142	30	98	20.6%	0.89 [0.51, 1.56]
	Lin, 2007 [33]	7	58	2	60	10.4%	3.98 [0.79, 20.03]
	Montes, 2009 [34]	40	180	78	352	21.8%	1.00 [0.65, 1.55]
	Melo, 2012 [37]	13	32	16	62	16.9%	1.97 [0.79, 4.87]
	Cosyn, 2016 [16]	10	28	1	28	7.3%	15.00 [1.76, 127.54]
	Saremi, 2021 [42]	20	100	7	178	17.0%	6.11 [2.48, 15.03]
Subtotal (95% CI)			574		822	100.0%	2.04 [1.02, 4.08]
Total events		131		136			
Heterogeneity: Tau^2^ = 0.53; Chi^2^ = 22.46, df = 6 (*p* = 0.0010); I^2^ = 73% Test for overall effect: Z = 2.01 (*p* = 0.04)							
TT vs. CC	Shimpuku, 2003 [29]	0	16	0	20		Not estimable
	Laine, 2006 [31]	4	39	5	29	27.7%	0.55 [0.13, 2.26]
	Lin, 2007 [33]	0	22	0	28		Not estimable
	Montes, 2009 [34]	2	54	8	114	25.7%	0.51 [0.10, 2.49]
	Melo, 2012 [37]	2	7	3	21	20.7%	2.40 [0.31, 18.55]
	Cosyn, 2016 [16]	2	8	0	13	12.3%	10.38 [0.43, 249.04]
	Saremi, 2021 [42]	4	38	0	82	13.6%	21.52 [1.13, 410.64]
Subtotal (95% CI)			184		307	100.0%	1.73 [0.45, 6.58]
Total events		14		16			
Heterogeneity: Tau^2^ = 1.16; Chi^2^ = 8.40, df = 4 (*p* = 0.08); I^2^ = 52% Test for overall effect: Z = 0.80 (*p* = 0.42)							
CT vs. CC	Shimpuku, 2003 [29]	1	17	2	22	3.5%	0.63 [0.05, 7.53]
	Laine, 2006 [31]	32	67	20	44	26.7%	1.10 [0.51, 2.35]
	Lin, 2007 [33]	7	29	2	30	3.2%	4.45 [0.84, 23.61]
	Montes, 2009 [34]	36	88	62	168	53.2%	1.18 [0.70, 2.01]
	Melo, 2012 [37]	9	14	10	28	5.0%	3.24 [0.85, 12.36]
	Cosyn, 2016 [16]	6	12	1	14	1.0%	13.00 [1.27, 133.28]
	Saremi, 2021 [42]	12	46	7	89	7.5%	4.13 [1.50, 11.40]
Subtotal (95% CI)			273		395	100.0%	1.68 [1.18, 2.39]
Total events		103		104			
Heterogeneity: Chi^2^ = 11.73, df = 6 (*p* = 0.07); I^2^ = 49% Test for overall effect: Z = 2.89 (*p* = 0.004)							
TT + CT vs. CC	Shimpuku, 2003 [29]	1	17	2	22	6.3%	0.63 [0.05, 7.53]
	Laine, 2006 [31]	36	71	25	49	20.6%	0.99 [0.48, 2.05]
	Lin, 2007 [33]	7	29	2	30	10.8%	4.45 [0.84, 23.61]
	Montes, 2009 [34]	38	90	70	176	23.1%	1.11 [0.66, 1.85]
	Melo, 2012 [37]	11	16	13	31	14.3%	3.05 [0.85, 10.90]
	Cosyn, 2016 [16]	8	14	1	14	7.1%	17.33 [1.75, 171.66]
	Saremi, 2021 [42]	16	50	7	89	17.7%	5.51 [2.08, 14.60]
Subtotal (95% CI)			287		411	100.0%	2.27 [1.11, 4.64]
Total events		117		120			
Heterogeneity: Tau^2^ = 0.51; Chi^2^ = 17.02, df = 6 (*p* = 0.009); I^2^ = 65% Test for overall effect: Z = 2.23 (*p* = 0.03)							
TT vs. CC + CT	Shimpuku, 2003 [29]	0	17	0	22		Not estimable
	Laine, 2006 [31]	4	71	5	49	41.6%	0.53 [0.13, 2.06]
	Lin, 2007 [33]	0	29	0	30		Not estimable
	Montes, 2009 [34]	2	90	8	176	39.5%	0.48 [0.10, 2.30]
	Melo, 2012 [37]	2	16	3	31	13.3%	1.33 [0.20, 8.92]
	Cosyn, 2016 [16]	2	14	0	14	3.1%	5.80 [0.25, 132.56]
	Saremi, 2021 [42]	4	50	0	89	2.5%	17.32 [0.91, 328.70]
Subtotal (95% CI)			287		411	100.0%	1.19 [0.58, 2.45]
Total events		14		16			
Heterogeneity: Chi^2^ = 6.85, df = 4 (*p* = 0.14); I^2^ = 42% Test for overall effect: Z = 0.47 (*p* = 0.63)							

Abbreviation: CI: confidence interval; I^2^: heterogeneity. Fixed-effects model was used for homozygous (CT vs. CC) and dominant (TT vs. CC plus CT) models.

**Table 5 pathogens-10-01600-t005:** Pooled analysis of the association between alleles and genotypes of *IL−1RN (VNTR)* polymorphism and the risk of dental PID.

Genetic Model	First Author, Publication Year	Case		Control		Weight	Odds Ratio
		Events	Total	Events	Total		M-H, Random, 95% CI
A2 vs. A1	Campos, 2005 [30]	19	54	17	66	47.5%	1.56 [0.71, 3.43]
	Petkovic-Curcin, 2017 [40]	18	68	50	128	52.5%	0.56 [0.29, 1.07]
Subtotal (95% CI)			122		194	100.0%	0.91 [0.34, 2.49]
Total events		37		67			
Heterogeneity: Tau^2^ = 0.39; Chi^2^ = 3.91, df = 1 (*p* = 0.05); I^2^ = 74% Test for overall effect: Z = 0.18 (*p* = 0.86)							
A2A2 vs. A1A1	Campos, 2005 [30]	3	14	2	20	53.4%	2.45 [0.35, 17.08]
	Petkovic-Curcin, 2017 [40]	0	22	11	36	46.6%	0.05 [0.00, 0.88]
Subtotal (95% CI)			36		56	100.0%	0.40 [0.01, 23.82]
Total events		3		13			
Heterogeneity: Tau^2^ = 7.19; Chi^2^ = 5.57, df = 1 (*p* = 0.02); I^2^ = 82% Test for overall effect: Z = 0.44 (*p* = 0.66)							
A1A2 vs. A1A1	Campos, 2005 [30]	13	24	13	31	46.7%	1.64 [0.56, 4.79]
	Petkovic-Curcin, 2017 [40]	12	34	28	53	53.3%	0.49 [0.20, 1.18]
Subtotal (95% CI)			58		84	100.0%	0.86 [0.26, 2.81]
Total events		25		41			
Heterogeneity: Tau^2^ = 0.48; Chi^2^ = 2.91, df = 1 (*p* = 0.09); I^2^ = 66% Test for overall effect: Z = 0.25 (*p* = 0.80)							
A2A2 + A1A2 vs. A1A1	Campos, 2005 [30]	16	27	15	33	48.1%	1.75 [0.62, 4.88]
	Petkovic-Curcin, 2017 [40]	13	34	39	64	51.9%	0.40 [0.17, 0.93]
Subtotal (95% CI)			61		97	100.0%	0.81 [0.19, 3.45]
Total events		29		54			
Heterogeneity: Tau^2^ = 0.86; Chi^2^ = 4.71, df = 1 (*p* = 0.03); I^2^ = 79% Test for overall effect: Z = 0.29 (*p* = 0.77)							
A2A2 vs. A1A1 + A1A2	Campos, 2005 [30]	3	27	2	33	18.7%	1.94 [0.30, 12.53]
	Petkovic-Curcin, 2017 [40]	3	34	11	64	81.3%	0.47 [0.12, 1.80]
Subtotal (95% CI)			61		97	100.0%	0.74 [0.26, 2.09]
Total events		6		13			
Heterogeneity: Chi^2^ = 1.47, df = 1 (*p* = 0.23); I^2^ = 32% Test for overall effect: Z = 0.57 (*p* = 0.57)							

Abbreviation: CI: confidence interval; I^2^: heterogeneity. Fixed-effects model was used for dominant (A2A2 vs. A1A1 plus A1A2) model.

**Table 6 pathogens-10-01600-t006:** Pooled analysis of the association between composite genotype of *IL−1A (−889)/IL−1B (+3953)* and *IL−1A (−889)/IL−1B (+3954)* and the risk of dental PID (genotype-positive vs. genotype-negative).

The Composite Genotype	First Author, Publication Year	Case		Control		Weight	Odds Ratio
		Events	Total	Events	Total		M-H, Fixed, 95% CI
*IL−1A (−889)* and *IL−1B (+3953)*	Rogers, 2002 [28]	9	19	11	31	20.7%	1.64 [0.51, 5.23]
	Campos, 2005 [30]	8	28	10	34	30.3%	0.96 [0.32, 2.89]
	Vaz, 2012 [38]	25	55	27	100	49.1%	2.25 [1.13, 4.49]
Total (95% CI)			102		165	100.0%	1.73 [1.03, 2.92]
Total events		42		48			
Heterogeneity: Chi^2^ = 1.67, df = 2 (*p* = 0.43); I^2^ = 0% Test for overall effect: Z = 2.08 (*p* = 0.04)							
*IL−1A (−889)* and *IL−1B (+3954)*	Laine, 2006 [31]	34	71	22	49	40.5%	1.13 [0.54, 2.34]
	Lachmann, 2007 [32]	6	11	8	18	28.1%	1.50 [0.33, 6.77]
	Hamdy, 2011 [36]	17	25	5	25	31.4%	8.50 [2.34, 30.91]
Total (95% CI)			107		92	100.0%	2.31 [0.65, 8.17]
Total events		57		35			
Heterogeneity: Tau^2^ = 0.89; Chi^2^ = 7.19, df = 2 (*p* = 0.03); I^2^ = 72% Test for overall effect: Z = 1.29 (*p* = 0.20)							

## Data Availability

No new data were created or analyzed in this study. Data sharing is not applicable to this article.

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
