# Peer review of "Association between Interleukin-1 Polymorphisms and Susceptibility to Dental Peri-Implant Disease: A Meta-Analysis"

_pathogens, 2021, doi:10.3390/pathogens10121600_

Round 1

Reviewer 1 Report

The submitted manuscript is a systematic review with meta-analysis that evaluated the association between IL-1 polymorphisms and peri-implant diseases. My main concerns relate to: (i) novelty: other reviews have been published in the past on this topic and it is not accurately discussed how this systematic review provides any significant updates compared to the existing ones; (ii) low number of titles obtained with the electronic search; (ii) mixed data on polymorphisms associated with diseases with different nature and etiology

  • Peri-implant diseases (PID) is a generic term that includes and cluster of different diseases and conditions. The Authors should clarify the different definitions of implant failure, marginal bone loss, peri-implant mucositis, and peri-implantitis. The authors should also stress the different etiologies known to trigger each one of these different diseases/conditions.

  • Page 4 line 157: “A total of 212 papers were retrieved; 112 duplicates and 61 irrelevant records were removed”. The search led to a fairly low number of articles to start. After removal of duplicates, the authors started their screening with only 100 articles. The Authors should consider modifying their search formula and run again the screening process.

  • As implant failure, marginal bone loss, and peri-implantitis are distinct entities, they are expected to be regulated by different genetic pathways. The authors are encouraged to provide analyses for each one of these diseases/conditions. If analysis is provided only for the generic group of PID, polymorphisms that are statistically significantly associated with one of these four conditions will remain undetected.

  • The Authors should discuss more about the plausibility of why the publication year was a significant predictor of the pooled results for gene polymorphism.

  • The manuscript currently includes 11 tables and 3 figures, for the high number of 14 total media. The authors should consider reducing the number of media and displaying only the main figures/tables that are most significative for the manuscript. The others could be uploaded as supplementary materials.

  • Other reviews have been published in the past on this topic. The Authors should stress more the differences and improvements they implemented in the current review compared to the previous ones.

  • The Authors are encouraged to present possible clinical applications of polymorphism screening for Personalized Medicine applied to the field of Dental implantology and/or Peri-Implantitis.

Author Response

We thank Reviewer #1 for the care devoted to thoroughly review the present manuscript. The comments and suggestions helped us to improve the quality of the manuscript. Please find attached the detailed point-by-point-response.

Again, thank you very much for all your kind efforts. 

Reviewer 2 Report

General comment: The present work has dealt with a current topic, important to pay attention in the field of Implantology! The manuscript has been English in general is correct, but there are several sentences that need to be rephrased and made clear for the reader. After the minor review recommended to the authors, this work can be further evaluated for publication. Please find the comments as follows:

Comments and suggestions:

- Where the authors have used "peri-implant disease" in the titles of the tables, they must write "PID" and not its long name.

- In point 4, you used "next" many times; it will be better to find other synonyms in the respective places.

Author Response

We thank Reviewer #2 for the care devoted to thoroughly review the present manuscript. The comments and suggestions helped us to improve the quality of the manuscript. Please find attached the detailed point-by-point-response.

Again, thank you very much for all your kind efforts. 

Round 2

Reviewer 1 Report

I thank the authors or implementing the requested modifications. The revised manuscript significantly improved in quality compared to the first submission.